# Kaempferol Interferes with Varicella-Zoster Virus Replication in Human Foreskin Fibroblasts

**DOI:** 10.3390/ph15121582

**Published:** 2022-12-19

**Authors:** Subin Park, Na-Eun Kim, Bang Ju Park, Hak Cheol Kwon, Yoon-Jae Song

**Affiliations:** 1Department of Life Science, Gachon University, Seongnam-si 13120, Republic of Korea; 2Department of Electronic Engineering, Gachon University, Seongnam-si 13120, Republic of Korea; 3Natural Product Informatics Research Center, Korea Institute of Science and Technology (KIST), Gangneung Institute, Gangneung 25451, Republic of Korea

**Keywords:** varicella-zoster virus, antiviral, kaempferol

## Abstract

Kaempferol, a natural flavonoid abundantly found in plants, is known to have pharmacological properties, such as anti-inflammatory and anti-cancer effects. In this study, we investigated the antiviral effects of kaempferol against a varicella-zoster virus (VZV) clinical isolate in vitro. We found that kaempferol significantly inhibited VZV replication without exhibiting cytotoxicity. Kaempferol exerted its antiviral effect at a similar stage of the VZV life cycle as acyclovir, which inhibits VZV DNA replication. Taken together, our results suggest that kaempferol inhibits VZV infection by blocking the DNA replication stage in the viral life cycle.

## 1. Introduction

Varicella-zoster virus (VZV), a member of the alpha herpesvirus subfamily, is spherically shaped with a diameter of 150–200 nm and consists of a nucleocapsid layer containing a double-stranded DNA (dsDNA) genome of ~125 kb, a tegument layer, and an envelope layer embedded with viral glycoproteins. Primary VZV infection through the respiratory mucosa causes varicella (chickenpox) and establishes latency in cranial nerve ganglia, dorsal root ganglia, and autonomic ganglia [1]. During latency, VZV can reactivate and cause herpes zoster [2,3].

More than 90% of the world’s population is infected with VZV, and one-third suffers from VZV reactivation [4]. The main symptom of varicella is a vesicular skin rash that appears on the trunk, head, and face. Although most symptoms are mild, they can be 25 times more severe in adults than in children. Varicella can also lead to complications such as bacterial superinfection of the skin, encephalitis, and pneumonia [5,6,7]. The main symptoms of herpes zoster are a dermatomal rash accompanied by pain and itching. There is also a mild rash that is easy to treat, but it is a problem because the rash is widespread, lasts for several weeks, and exhibits a high frequency of reactivation in the elderly [1,8]. In addition, herpes zoster causes complications, especially neurological pain, and persistent pain can lead to post-herpetic neuralgia [9].

The development of antiviral therapies against VZV is important because VZV diseases continue to occur even in countries where VZV vaccines are routinely available [1] and because older, immunocompromised patients and those with chronic disease have a risk of developing complications [10,11]. The nucleoside analog acyclovir, a DNA chain terminator, was introduced as a specific antiviral drug for VZV in the early 1980s [10,11] however, its poor absorption and serious nephrotoxicity led to the subsequent development of valacyclovir and famciclovir [12,13,14]. Despite the availability of these therapeutic agents, next-generation antiviral therapies with different mechanisms of action are required due to the development of resistance to nucleoside analogs. The development of antiviral drugs with higher efficacy is also desirable because such agents would help reduce nerve damage and complications through rapid treatment [15,16].

Flavonoids such as kaempferol, which is known to have pharmacological properties—including anti-inflammatory and anticancer effects [17,18] are abundant in plants [19]. Kaempferol also inhibits infection of African swine fever virus, Influenza virus, Japanese encephalitis virus, and Herpes simplex virus [20,21,22,23]. Thus, in this study, we investigated the inhibitory effects of kaempferol on VZV infection and its mechanism of action in vitro.

## 2. Results

### 2.1. Kaempferol Inhibits VZV Replication

The effect of kaempferol on VZV replication was determined by conducting plaque reduction assays. To this end, human foreskin fibroblasts (HFFs) were inoculated with the cell-associated clinical VZV isolate, YC01 (VZV-YC01), at a multiplicity of infection (MOI) of 0.1 and then treated with kaempferol at concentrations of 5, 7, 10, and 15 µg/mL or with acyclovir at concentrations of 0.5, 1, 5, and 10 µM; dimethyl sulfoxide (DMSO) was used as a vehicle control. Cells were re-treated with kaempferol, acyclovir or DMSO on day 3 after inoculation and the number of plaques was counted on day 6. Concentration-response studies showed that kaempferol at a concentration of 7 µg/mL or higher suppressed VZV replication by more than 50% compared with cells treated with DMSO and yielded an estimated 50% inhibitory concentration (IC50) value of 6.36 ± 0.73 µg/mL (Figure 1). The IC50 value of acyclovir, which was used as a reference control, was determined to be 0.54 ± 0.12 µM (Appendix A).

To assess the cytotoxicity of kaempferol, we treated HFFs with kaempferol (5, 10, 15, and 20 µg/mL) or DMSO for 72 h, then determined the cytotoxicity using Thiazolyl Blue Tetrazolium Bromide (MTT) assays. Kaempferol caused no cytotoxicity at any concentration tested (Figure 2). Collectively, these results indicate that kaempferol inhibits VZV replication without exhibiting cytotoxicity.

### 2.2. Kaempferol Has No Effect on Expression of VZV Immediate-Early Genes

During VZV replication, immediate early (IE), early (E), and late (L) genes are expressed in order, starting with the expression of IE62, a major transactivator protein [2,24]. We first determined whether IE62 promoter activity is affected by kaempferol by transfecting 293T cells with a vector expressing firefly luciferase gene under control of the VZV IE promoter and then treating cells with kaempferol, acyclovir (ACV), or DMSO 6 h later. Luciferase activity was measured after 24 h (Figure 3). Even at 10 µg/mL, a concentration that inhibited VZV replication by more than 70%, kaempferol had no effect on IE promoter activity compared with DMSO. Acyclovir, which blocks viral DNA synthesis [11], also had no effect on IE promoter activity. These results indicate that kaempferol interferes with the VZV life cycle at a stage after IE gene expression.

### 2.3. Determination of the Time Point at Which Kaempferol Suppresses VZV Infection

To determine the stage of the VZV life cycle inhibited by kaempferol, we performed a time-of-drug-addition assay using acyclovir as a control (Figure 4A). HFFs were pre-treated with kaempferol, acyclovir, or DMSO for 3 h and inoculated with VZV-YC01 at a MOI of 0.1. After virus inoculation for 1 h, drugs were removed, and HFFs were incubated (pre-treatment). In addition, HFFs were inoculated with VZV-YC01 at a MOI of 0.1 and treated with kaempferol, acyclovir, or DMSO at 0, 6, 12, 18, 24, or 36 h after inoculation. At 72 h after inoculation, the relative amount of viral DNA was determined by quantitative polymerase chain reaction (qPCR) (Figure 4B). Pre-treatment with acyclovir or kaempferol had no effect on VZV replication, indicating that kaempferol does not affect viral attachment to cells or viral integrity. Acyclovir inhibited VZV replication by 94% at 12 h, an inhibitory effect that was reduced to 89.3% in cells treated with acyclovir at 18 h and was further significantly reduced in cells treated at 24 h after inoculation. Similar to acyclovir, kaempferol inhibited VZV replication by 62% in cells treated at 12 h after inoculation and by 56.5% at 18 h. The inhibitory effect of kaempferol on VZV replication was significantly reduced (to 28.6%) in cells treated with kaempferol at 24 h after inoculation. These results suggest that kaempferol interferes with the VZV life cycle at a similar stage as acyclovir.

### 2.4. Kaempferol Inhibits VZV DNA Replication

To determine whether kaempferol targets the DNA synthesis stage of the VZV life cycle, we first determined the time point at which VZV DNA synthesis starts. Since one VZV life cycle takes 18–22 h [25], we harvested HFFs at 0, 3, 6, 9, 12, 18, and 24 h after VZV-YC01 infection and analyzed the amount of VZV DNA by qPCR (Figure 5A). Under our experimental conditions, VZV DNA synthesis was initiated between 9 and 12 h after inoculation (Figure 5A). To determine the inhibitory effect of kaempferol against VZV DNA synthesis, we inoculated cells with VZV-YC01 and then treated them with kaempferol, acyclovir, or DMSO. We then harvested cells at 6, 12, 18, and 24 h after VZV inoculation, extracted total DNA, and determined the amount of viral DNA by qPCR (Figure 5B). Only acyclovir significantly inhibited VZV DNA synthesis at 12 h after inoculation. However, both kaempferol and acyclovir significantly inhibited VZV DNA synthesis at 18 h after inoculation. These results indicate that kaempferol interferes with VZV DNA synthesis at later time points compared with acyclovir.

## 3. Discussion

Kaempferol is abundant in medicinal plants such as *Eugenia jambolana* Lam (1.3 mg/kg), *Acacia nilotica* L. (21.7 mg/kg), *Azadirachta indica* A. Juss. (0.5 mg/kg), *Terminalia arjuna* (Roxb.) Wight & Arn. (8.9 mg/kg), *Ficus religiosa* L. (160.8 mg/kg), *Aloe barbadensis* Miller (257.7 mg/kg), *Rosmarinus officinalis* L. (1.2 mg/kg), and *Euonymus alatus* Thunb. (5 mg/kg) [19,26]. Furthermore, various dietary sources, including kale and spinach, contain significant amounts of kaempferol (mg/100 g fresh weight) [27].

We found that kaempferol possesses potent antiviral activity against VZV without exhibiting cytotoxicity (Figure 1 and Figure 2). Thus, we further determined the stage of the VZV life cycle targeted by kaempferol. Kaempferol had no effect on the VZV entry into host cells or the activity of the VZV IE promoter (Figure 3 and Figure 4). Interestingly, kaempferol interfered with VZV DNA synthesis, and a time-of-drug-addition assay indicated that kaempferol inhibits VZV replication at similar time points as acyclovir (Figure 4 and Figure 5). Acyclovir is mono-phosphorylated by viral thymidine kinase (TK) and further phosphorylated by cellular TKs into acyclovir triphosphate, which is an analog of deoxyguanosine triphosphate (dGTP). Acyclovir triphosphate, which cannot form a phosphodiester bond with the next nucleotide, competes with dGTP as a substrate for viral DNA polymerase and terminates viral DNA synthesis. Since kaempferol did not affect VZV IE, E, or L gene expression (Figure 3 and Appendix A), it might interfere with the process of VZV DNA synthesis by blocking viral DNA polymerase activity and/or cellular factors essential for viral DNA replication.

Previous studies have reported antiviral activities of kaempferol, showing that kaempferol functions as a neuraminidase inhibitor for influenza viruses [21] and a reverse transcriptase inhibitor for human immunodeficiency virus I (HIV-1) [28]. Furthermore, kaempferol inhibits Japanese encephalitis virus (JEV) by acting directly on JEV RNA [22] and inhibits the entry and post-entry stages of African swine fever virus. The post-entry stage of the virus life cycle is associated with autophagy induction, and kaempferol causes cell cycle arrest and impacts autophagic processes [20,29]. VZV infections are suppressed by inhibition of autophagy [30]. Thus, kaempferol interferes with VZV replication by inhibiting autophagy in addition to blocking viral DNA synthesis. Although further studies are required to elucidate its mechanism of action and increase its antiviral potency, kaempferol is a potential candidate for a universal antiviral agent.

## 4. Materials and Methods

### 4.1. Cells, Viruses, and Materials

Human foreskin fibroblasts (HFFs) and HEK293T cells were cultured in Dulbecco’s modified Eagle’s medium (Hyclone, Logan, UT, USA) supplemented with 10% fetal bovine serum and 1X penicillin-streptomycin at 37 °C in a humidified 5% CO_2_ environment. The maintenance and propagation of the clinical VZV isolate, YC01 (VZV-YC01), have been described previously [31]. Kaempferol and acyclovir were purchased from Sigma-Aldrich (St. Louis, MO, USA).

### 4.2. Plaque Reduction Assay

HFFs in a 12-well plate were inoculated with serially diluted cell-associated VZV-YC01. At 1 h after inoculation, cells were treated with kaempferol (5, 7, 10, and 15 µg/mL), acyclovir (0.5, 1, 5, and 10 µM), or DMSO (vehicle control), and subsequently re-treated with kaempferol or acyclovir at the same concentration (or DMSO) after 3 days. At 6 days after inoculation, cells were fixed by incubating with 10% formaldehyde for 10 min at room temperature and then stained with 0.3% crystal violet. The next day, the number of plaques was counted and the half-maximal inhibitory concentration (IC50) value was calculated using GraphPad Prism 7 (GraphPad Software, San Diego, CA, USA).

### 4.3. MTT Assay

HFFs in a 96-well plate were treated with kaempferol at 5, 10, 15, and 20 µg/mL, or with DMSO. After 48 h, cell viability was measured using MTT assays as described previously [32]. MTT was purchased from Sigma-Aldrich.

### 4.4. Plasmids, Transfections, and Luciferase Reporter Assays

HCMV MIE enhancer/promoter sequences were removed from the vector pJHA324 using the restriction enzyme *Hind*III (Enzynomics, Daejeon, Korea) [33]. The VZV IE promoter was synthesized (Macrogen, Seoul, Korea) and amplified by PCR using a primer containing the *Hind*III sequence at the end. The primer sequences used for amplification were 5′-CCC AAG CTT ATC GTC TGT AGA CAC ACG ATG-3′ (forward) and 5′-CCC AAG CTT CGC ACT GGG GTG AAT TTA G-3′ (reverse). The PCR product was digested with *Hind*III and ligated using T4 ligase (Enzynomics) into a pJHA324 vector in which the HCMV MIE enhancer/promoter sequences had been removed. The orientation of the VZV IE promoter insert was confirmed by DNA sequencing (Macrogen, Seoul, Korea). Transient transfections and luciferase assays were performed using Omicsfect™ (Omics Bio, Taipei city, Taiwan) and Dual-Luciferase Reporter Assay System (Promega, Madison, WI, USA), respectively, according to the manufacturers’ protocols as described previously [34].

### 4.5. Quantitative Polymerase Chain Reaction (qPCR)

For quantitative analysis of viral DNA, nuclei of VZV-infected cells were prepared as described previously [25], and total DNA was isolated using an AccuPrep Genomic DNA Extraction kit (Bioneer, Daejeon, Republic of Korea). Viral DNA was quantified by qPCR according to a previous report [35]. For quantitative analysis of viral mRNA, quantitative reverse transcription PCR (qRT-PCR) were performed as described previously [36]. The primer sequences used for amplification were as follows: VZV ORF62, 5′-TCTTGTCGAGGAGGCTTCTG-3′ (forward) and 5′-TGTGTGTCCACCGGATGAT-3′ (reverse); VZV ORF28 (E), 5′-CGAACACGTTCCCCATCAA-3′ (forward) and 5′-CCCGGCTTTGTTAGTTTTGG-3′ (reverse); VZV gB (L), 5′-GATGGTGCATACAGAGAACATTCC-3′ (forward) and 5′-CCGTTAAATGAGGCGTGACTAA-3′; GAPDH, 5′-CATGAGAAGTATGACAACAGCCT-3′ (forward) and 5′-AGTCCTTCCACGATACCAAAGT- 3′ (reverse).

### 4.6. Statistical Analysis

Data are presented as means ± standard deviations (SD). The significance of differences between means was determined with the Student’s *t*-test. *p*-values were determined by unpaired two-tailed Student’s *t*-tests. *p*-values <0.05 were considered statistically significant.

## Figures and Tables

**Figure 1 pharmaceuticals-15-01582-f001:**
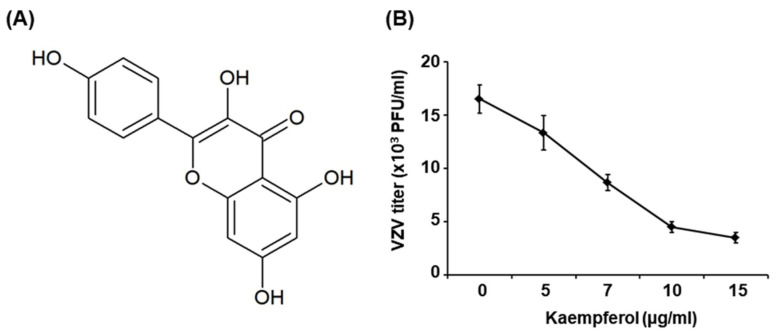
Anti-VZV activity of kaempferol. (**A**) Structure of kaempferol. (**B**) HFFs were inoculated with serially diluted cell-associated VZV-YC01 and treated with kaempferol at the indicated concentrations. Cells were re-treated with the same concentrations of kaempferol 3 days after inoculation. At 6 days after inoculation, cells were stained with 0.3% crystal violet and the number of plaques was counted.

**Figure 2 pharmaceuticals-15-01582-f002:**
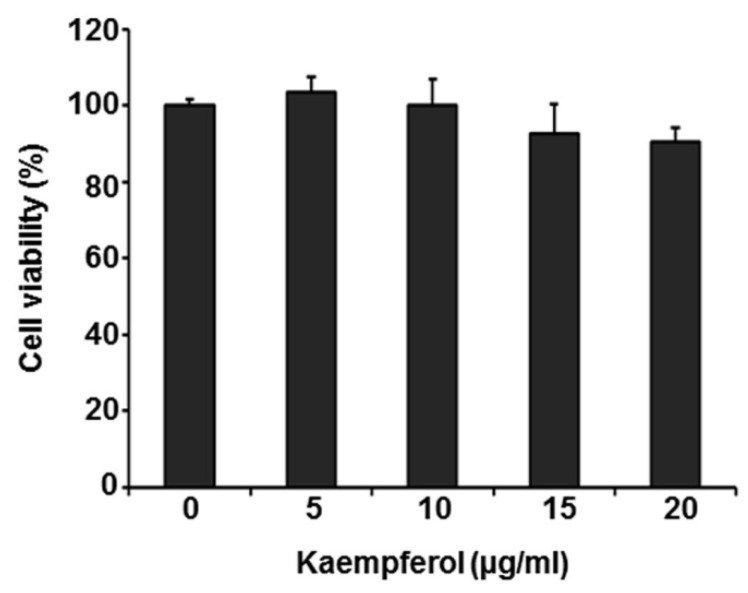
Cytotoxicity of kaempferol. HFFs were treated with kaempferol at the indicated concentrations. At 72 h after treatment, cell viability was determined by MTT assay and expressed relative to that of cells treated with DMSO (defined as 100%).

**Figure 3 pharmaceuticals-15-01582-f003:**
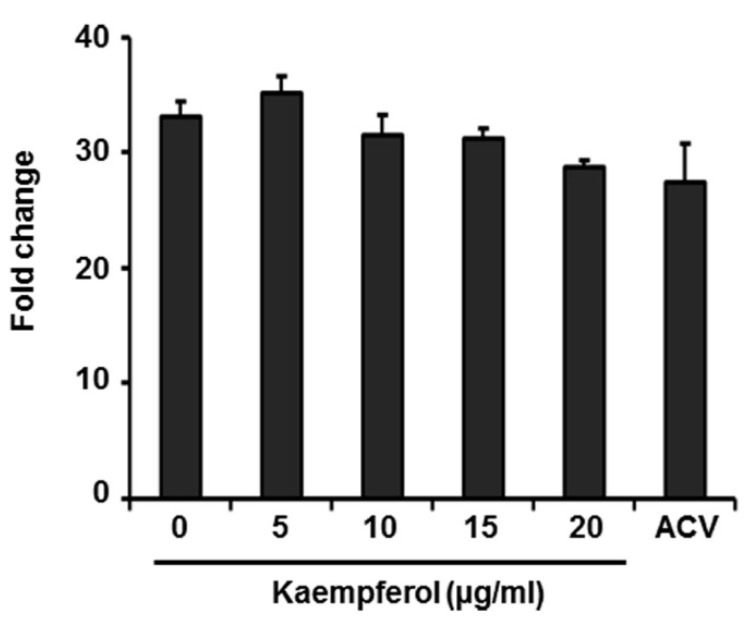
Effect of kaempferol on VZV IE promoter activity. HEK293T cells were transfected with VZV IE promoter-driven firefly luciferase plus control *Renilla* luciferase plasmids. Cells were treated with kaempferol, acyclovir (ACV), or DMSO at 6 h after transfection, and promoter activity was analyzed using a dual luciferase assay at 18 h after treatment. Relative luciferase activity was calculated as VZV IE promoter-driven firefly luciferase activity relative to that in cells transfected with the control vector (defined as 1). Data represent the average of three independent experiments.

**Figure 4 pharmaceuticals-15-01582-f004:**
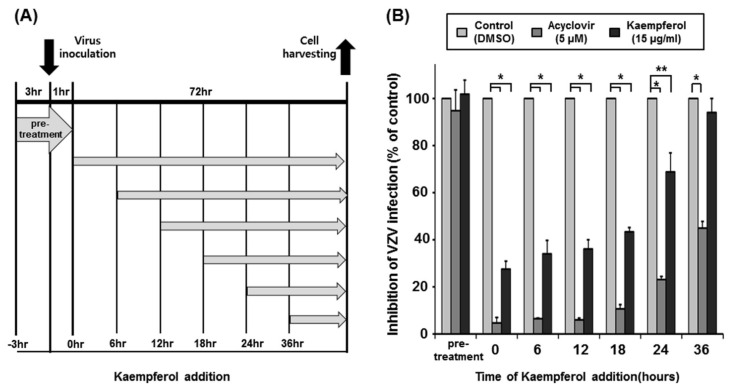
Time-of-drug-addition assay to identify the target of kaempferol. (**A**) Schematic diagram of the assay. (**B**) HFFs were pre-treated with kaempferol, acyclovir, or DMSO for 3 h and inoculated with VZV-YC01 at a MOI of 0.1. After virus inoculation for 1 h, drugs were removed, and HFFs were incubated for 72 h (pre-treatment). In addition, HFFs were inoculated with cell-associated VZV-YC01 at a MOI of 0.1 and treated with kaempferol (15 µg/mL), acyclovir (5 µM), or DMSO at 0, 6, 12, 18, 24, and 36 h after inoculation. Cells were harvested 72 h after inoculation, at which point total DNA was extracted and the relative amount of viral DNA was analyzed by qPCR using primers specific for open reading frame 62 (ORF62) and glyceraldehyde 3-phosphate dehydrogenase (GAPDH). Experiments were performed in triplicate and GAPDH-normalized values were expressed relative to those in cells treated with DMSO (defined as 100%). * *p* < 0.005, ** *p* < 0.05 (Student’s *t*-test).

**Figure 5 pharmaceuticals-15-01582-f005:**
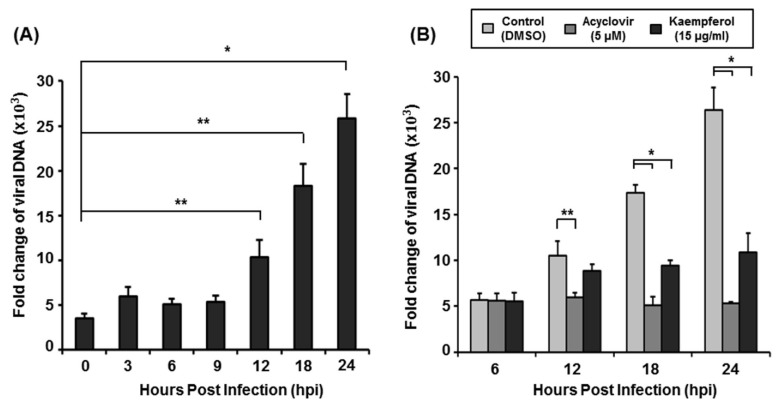
Analysis of the amount of VZV DNA replication. (**A**) HFFs were inoculated with cell-associated VZV-YC01 at a MOI of 0.1 and harvested at 0, 3, 6, 9, 12, 18, and 24 h. (**B**) VZV-YC01–inoculated HFFs were treated with kaempferol (15 µg/mL), acyclovir (5 µM), or DMSO, and harvested at 0, 6, 12, 18, and 24 h. Total DNA was extracted from nuclei of VZV-infected cells and relative amounts of viral DNA were determined by qPCR as above. Experiments were performed in triplicate, and values were expressed relative to those in uninfected cells (defined as 1). * *p* < 0.005, ** *p* < 0.05 (Student’s *t*-test).

## Data Availability

Data is contained within the article and Appendix A.

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
