# Peer review of "Kaempferol Interferes with Varicella-Zoster Virus Replication in Human Foreskin Fibroblasts"

_pharmaceuticals, 2022, doi:10.3390/ph15121582_

Round 1

Reviewer 1 Report

Because a link between metabolism and toxicity was not clearly identified trough the  phases the MTK’s toxicity is likely mediated via  undefined interactions that rest to discover. At moment data could be not conclusive altought the experimental work is good.

Author Response

Because a link between metabolism and toxicity was not clearly identified through the phases the MTT’s toxicity is likely mediated via undefined interactions that rest to discover. At moment data could be not conclusive although the experimental work is good.

- As addressed by the reviewer, the link between metabolism and toxicity has not been clearly defined. However, MTT assay is generally used to evaluate cell toxicity by measuring the activity of mitochondria in cells.

Reviewer 2 Report

In my point of view, the manuscript is suitable for publication. I have a few recommendations concerning “Materials and methods”

1.      line 158- 4.2. Plaque reduction assay - Please specify the kaempferol concentrations

2.      Despite the figure legends consist how the statistical analysis was performed, I believe it would be more clear if a subsection “Statistical analysis” is added to Materials and methods

Author Response

1. line 158- 4.2. Plaque reduction assay - Please specify the kaempferol concentrations.

- As suggested by the reviewer, the concentrations are included in the materials and methods section (lines 207-208).

2. Despite the figure legends consist how the statistical analysis was performed, I believe it would be more clear if a subsection “Statistical analysis” is added to Materials and methods.

- As suggested by the reviewer, statistical analysis is included in the materials and methods section (lines 244-247).

Reviewer 3 Report

The manuscript “Kaempferol interferes with varicella-zoster virus replication in human foreskin fibroblasts” is devoted to investigation the inhibitory effects of kaempferol on varicella-zoster virus and its mechanism of action in vitro.

I have the following comments to the authors:

-          Authors need to resize the Figures, they are too big.

-          According to the Instructions for Authors all Figures should be inserted into the main text close to their first citation.

-          It is not clear from the Introduction why the authors chose Eonymus alatus as the source of kaempferol.

-          The authors refer to the screening results in Figure S1. However, Figure S1 shows only the antiviral activity of Eonymus alatus. Why are the rest of the objects not shown?

-          There should be no experimental data in the Introduction (Table S1). Experimental data should be indicated in the Results section.

-          There is no reference compound in Figure S1. How to interpret the results?

-          Authors said that “Flavonoids such as kaempferol … are abundant in Euonymus alatus (Thunb.) [19,20].” Authors refer to 2 articles [19, 20] that do not mention the quantitative assessment of kaempferol in Eonymus alatus. On what basis were such conclusions drawn?

-          There is no reference compound in Figure 1. How to interpret the results?

-          All Figures, in addition to captions, also contain a description of the experiments. Description of experiments should be in Materials and methods.

-          Line 116. Provide data on screening studies.

-          Lines 152-153. There is no data on obtaining the extract. There is no detailed description of obtaining the extract, its quantitative characteristics.

I cannot recommend the manuscript for publication in a highly scientific journal of the 1st quartile Pharmaceuticals.

Author Response

1. It is not clear from the Introduction why the authors chose Eonymus alatus as the source of kaempferol.

- As indicated in line 49-56, our previous study using 70% ethanol extracts of 662 plants found that the Euonymus alatus (Thunb) (EAE) extract inhibited VZV replication (Reference #17). Since kaempferol is the most abundant flavonoid in EAE, we investigated the antiviral activity of kaempferol against VZV in this study.  

2. The authors refer to the screening results in Figure S1. However, Figure S1 shows only the antiviral activity of Eonymus alatus. Why are the rest of the objects not shown?

- As addressed above, the screening using 662 plant extracts was performed in the previous study (Reference #17). In this study, we focused on the anti-VZV activity of kaempferol which is the most abundant flavonoid in ESE.

3. There should be no experimental data in the Introduction (Table S1). Experimental data should be indicated in the Results section.

- As suggested by the reviewer, Figure S1 is included in the results section (lines 59-60).

4. There is no reference compound in Figure S1. How to interpret the results?

- Figure S1 presents anti-VZV activity of EAE and its IC50 value. The number of plaques formed decreased in a dose-dependent manner, and the IC50 was defined by the concentration of EAE required for 50% reduction in plaque numbers. A reference compound could be acyclovir used in Figures 3, 4 and 5. The IC50 of acyclovir against VZV p-Oka is 13 ± 3 µM as previously reported (Morfin, F., Thouvenot, D., De Turenne-Tessier, M., Lina, B., Aymard, M., and Ooka, T. 1999. Phenotypic and genetic characterization of thymidine kinase from clinical strains of varicellazoster virus resistant to acyclovir. Antimicrob. Agents Chemother. 43, 2412–2416).

5. Authors said that “Flavonoids such as kaempferol … are abundant in Euonymus alatus (Thunb.) [19,20].” Authors refer to 2 articles [19, 20] that do not mention the quantitative assessment of kaempferol in Eonymus alatus. On what basis were such conclusions drawn?

- In reference #20, “flavonoids are the most abundant and major bioactive constituents in E. alatus.” is stated. In references #21 and 22, it is quantitatively confirmed that flavonoids are contained in relatively large amounts among the active substances extracted from the Euonymus alatus. In particular, kaempferol shows the highest content among active substances at 5 µg /g. These information is included in the manuscript (lines 50-53, 59-62)

6. There is no reference compound in Figure 1. How to interpret the results?

- Figure 1 presents anti-VZV activity of kaempferol and its IC50 value. The number of plaques formed decreased in a dose-dependent manner, and the IC50 was defined by the concentration of kaempferol required for 50% reduction in plaque numbers. A reference compound could be acyclovir used in Figures 3, 4 and 5. The IC50 of acyclovir against VZV p-Oka is 13 ± 3 µM as previously reported (Morfin, F., Thouvenot, D., De Turenne-Tessier, M., Lina, B., Aymard, M., and Ooka, T. 1999. Phenotypic and genetic characterization of thymidine kinase from clinical strains of varicellazoster virus resistant to acyclovir. Antimicrob. Agents Chemother. 43, 2412–2416).

7. All Figures, in addition to captions, also contain a description of the experiments. Description of experiments should be in Materials and methods.

- To assist the readers understand figures, experimental procedures are briefly described in figure legends.

8. Line 116. Provide data on screening studies.

- As addressed above, the screening using 662 plant extracts was performed in the previous study (Reference #17). In this study, we focused on the anti-VZV activity of kaempferol which is the most abundant flavonoid in ESE.

9. Lines 152-153. There is no data on obtaining the extract. There is no detailed description of obtaining the extract, its quantitative characteristics.

- As suggested by the reviewer, the information is included in the materials and methods section (lines 202-204).

Reviewer 4 Report

The authors studied the antiviral effect of kaempferol against a varicella-zoster virus (VZV) in vitro. They found that kaempferol significantly inhibited VZV replication without exhibiting citotoxicity. They suggested that kaempferol inhibits VZV infection by blocking the DNA replication stage in the viral life cycle.

The manuscript is suitable for publication after minor revision.

The abbreviations should be explained where they are mentioned at first, eg: line 91. qPCR, line 334. ORF62, GAPDH.

Line 120.: One reference is missing.

Author Response

1. The abbreviations should be explained where they are mentioned at first, eg: line 91. qPCR, line 334. ORF62, GAPDH.

- The manuscript has been revised as suggested by the reviewer (lines 78-79, 112-113, 130-131).

2. Line 120: One reference is missing.

- A reference is included as suggested by the reviewer (line 162).

Reviewer 5 Report

This paper is interesting and may be considered for publication. However, there are several points should be revised:

1. Discuss about contents of kaempferol in different plants

2. I need you provide your preliminary experiments on screening many plants, that lead to the reasons you conduct this research. Please provide details of this experiment and place in this paper.

3. Discuss about mechanism of antiviral activities of kaempferol which have been clarified so far, do the mechanism on your described VZA is new? Please compared the known mechanism of kaempferol on other viruses which have been described so far.

Author Response

1. Discuss about contents of kaempferol in different plants

- As suggested by the reviewer, the plant and dietary sources of kaempferol are discussed in lines 163-168.

2. I need you provide your preliminary experiments on screening many plants, that lead to the reasons you conduct this research. Please provide details of this experiment and place in this paper.

- As indicated in line 49-56, our previous study using 70% ethanol extracts of 662 plants found that the Euonymus alatus (Thunb) (EAE) extract inhibited VZV replication (Reference #17). Since kaempferol is the most abundant flavonoid in EAE, we investigated the antiviral activity of kaempferol against VZV in this study.  

3. Discuss about mechanism of antiviral activities of kaempferol which have been clarified so far, do the mechanism on your described VZA is new? Please compared the known mechanism of kaempferol on other viruses which have been described so far.

- This is the first study reporting that kaempferol inhibits VZV DNA synthesis. The antiviral mechanisms of kaempferol against other viruses are discussed in lines 183-191.

Round 2

Reviewer 3 Report

The authors did not answer some important questions.

The authors mention the results of screening several times in the study, in particular, in the Discussion part, which implies the presentation of the results of the experiment. Screening results are still not shown in Figure S1.

Also, information about the reference substance in Figures S1 and 1 has not been added. The authors mention that the reference compound "may be acyclovir". Add missing information.

There is no detailed description of obtaining the extract, its quantitative characteristics. The authors gave a link to another plant object. Quantitative characterization of Chrysanthemum indicum extract cannot match the quantitative characterization of Euonymus alatus extract.

Author Response

1. The authors mention the results of screening several times in the study, in particular, in the Discussion part, which implies the presentation of the results of the experiment. Screening results are still not shown in Figure S1.

- The main subject of this manuscript is the antiviral activity of kaempferol, not ethanol extract of  E. alatus, against VZV. Initially, we described the previous screening results to explain one of reasons why we investigated antiviral activities of kaempferol. Since the data with EAE disorient the main subject of this manuscript, we have revised the manuscript to excluded them (lines 49-52, 58-64, 159-163). 

2. Also, information about the reference substance in Figures S1 and 1 has not been added. The authors mention that the reference compound "may be acyclovir". Add missing information.

- As suggested by the reviewer, the data with acyclovir are included as Figure S1 (lines 69-71).

3. There is no detailed description of obtaining the extract, its quantitative characteristics. The authors gave a link to another plant object. Quantitative characterization of Chrysanthemum indicum extract cannot match the quantitative characterization of Euonymus alatus extract.

- The same method was used as described in the reference paper. The dried plant material (1.2 kg) was exhaustively extracted by 70% EtOH, and the solvent was evaporated under reduced pressure at below 40C. As answered above,  the data with EAE is removed from the manuscript to improve readability. Hence, the method section has been revised (lines 202-205, 209-211).